# A Single Bout of Foam Rolling After Nordic Hamstring Exercise Improves Flexibility but Has No Effect on Muscle Stiffness or Functional Muscle Parameters

**DOI:** 10.3390/medicina61081486

**Published:** 2025-08-19

**Authors:** Coşkun Rodoplu, Christian Burger, Josef Fischer, Josefina Manieu Seguel, Ramiz Arabacı, Andreas Konrad

**Affiliations:** 1Department of Physical Education and Sports Teaching, Bursa Uludağ University, Bursa 16059, Turkey; coskun.rodoplu@btu.edu.tr (C.R.); ramizar@uludag.edu.tr (R.A.); 2Department of Common Courses, Bursa Technical University, Bursa 16310, Turkey; 3Institute of Human Movement Science, Sport and Health, Graz University, 8010 Graz, Austria; christian.burger@uni-graz.at (C.B.); josef.fischer@uni-graz.at (J.F.); josefina.manieu@uni-graz.at (J.M.S.)

**Keywords:** eccentric exercise, range of motion, recovery, muscle performance

## Abstract

*Background and Objectives*: The Nordic hamstring exercise (NHE) effectively strengthens the hamstrings, reduces the risk of hamstring strain, and induces fatigue in the muscles; thus, post-NHE recovery strategies should be optimized. Foam rolling (FR) is a widely used method, with the belief that it can speed up recovery. Thus, this study investigated the acute and 48-h effects of FR following the NHE on muscle stiffness, pain pressure threshold (PPT), flexibility, countermovement jump (CmJ) height, and maximal voluntary isometric contraction (MVIC). *Materials and Methods*: Thirty-two active males were randomly assigned to either an FR group (*n* = 16) or a passive recovery (PR, *n* = 16) group. Measurements were taken at three time points: pre-test, post-test_0h, and post-test_48h. Participants performed the NHE (3 sets × 10 reps) before the recovery interventions. Variables assessed included muscle stiffness (MyotonPro), flexibility (sit and reach), PPT (algometer), jump performance (force platform), and MVIC peak torque (dynamometer). *Results*: FR significantly improved acute flexibility (12.7%, *p* < 0.001) following the NHE and prevented flexibility loss at 48 h, compared to the PR group. However, FR showed no significant advantages over PR in terms of muscle stiffness, PPT, CmJ, or MVIC, both acutely and at 48 h (*p* > 0.05). *Conclusions*: FR is an effective acute recovery strategy for improving flexibility after the NHE but offers limited effects for muscle stiffness, PPT, and other functional muscle parameters, both acutely and at 48 h. Further research should explore the long-term effects and efficacy of FR across diverse populations and recovery scenarios.

## 1. Introduction

A single bout of the Nordic hamstring exercise (NHE) can develop high maximal eccentric hamstring strength torque and, when performed over several weeks, can prevent hamstring injuries and improve long-term muscle performance [1,2]. The NHE can reduce the initial and recurrent injury rates in professional football players by 60 and 85%, respectively, while also improving eccentric strength, flexibility, and jump height [1]. The mechanism behind this is not fully understood, but the NHE induces mechanical stress and micro-tears in muscle fibers [3]. In addition, as the NHE acutely increases muscle stiffness due to acute cell swelling, it can cause injury if not properly managed [4,5]. In this context, a better understanding of the muscle structures and recovery processes after the NHE may help prevent injuries and optimize performance [4].

Fatigue refers to a temporary decline in muscle function, often caused by physical overload, and is characterized by increased stiffness, reduced flexibility, and lower force production (e.g., decreased MVIC) [6,7]. Post-exercise recovery plays a crucial role in enhancing performance and reducing fatigue [8]. Athletes experience physiological stress due to overload during exercise or competition, which results in fatigue that can impair performance [6]. During recovery from muscle fatigue, changes in muscle structure and mechanics occur, particularly in flexibility, pain, and stiffness, which seem to be critical for recovery [9,10]. Overload-related fatigue during exercise can be minimized with proper recovery interventions [7,11].

Passive recovery (PR), stretching exercises, hot/cold water immersion, and massage are commonly used methods for post-exercise acute recovery [10,12]. In recent years, foam rolling (FR) interventions have become increasingly popular for use before and after exercise [11,13,14]. FR may be a distinctly effective and practical tool compared with these modalities to accelerate recovery by increasing blood flow, reducing tissue stiffness through thixotropic effects, modulating pain via mechanoreceptor stimulation, and alleviating muscle soreness, although the precise mechanisms remain unclear [10,15,16]. However, while various studies have supported the benefits of FR [9,10,17], a limited number of studies have presented opposing views. This could be due to the incomplete understanding of the neurological and morphological mechanisms underlying the effects of FR [15,16]. The effects of FR on muscle function (e.g., contraction, flexibility, pain pressure threshold [PPT], and stiffness) and performance (e.g., jump performance) after resistance exercises remain a focus of ongoing research [7,11].

Recent studies have examined the chronic and acute effects of FR interventions following eccentric exercise [13,18,19]. Some studies have indicated that acute FR effects improve muscle function, performance, and range of motion (ROM) [13,19]. In addition, FR has been shown to immediately increase ROM post-exercise and to sustain this effect for up to 30 min [11,20]. In contrast, other studies have suggested that FR has no acute effect on ROM [21,22]. However, it has been shown that chronic FR can reduce muscle stiffness, increase flexibility, and lower the risk of sports injuries [23,24]. There is also evidence that FR can accelerate muscle repair and promote faster healing of micro-tears. It is commonly accepted that a single FR intervention can be used for muscle preparation before and after exercise and, in the long term, to prevent chronic issues such as injuries and improve performance [11,18,23].

To date, although there have been a few studies on the effects of FR on hamstring muscle groups in recovery strategies [25,26], no study has compared the effects of FR applied after the NHE on functional (flexibility, jump performance, maximum voluntary isometric contraction [MVIC] torque), structural (muscle stiffness), and neurological parameters (PPT). The aim of the present study was to investigate the immediate acute effects and sustained effects (48 h) of an FR intervention on muscle stiffness, PPT, flexibility, jump performance, and MVIC (peak power) after a fatiguing NHE. Based on previous studies, we hypothesized that the FR intervention, both immediately after the NHE and 48 h later, would have a greater effect than passive recovery (PR) in increasing flexibility, elevating PPT, decreasing muscle stiffness, enhancing MVIC peak torque, and in jump performance.

## 2. Materials and Methods

### 2.1. Participants

The study included 32 healthy males aged 18–30. Based on a similar study by Nakamura et al. [19], for the sample size calculation, *f* = 0.4, α = 0.05 (5% probability of type 1 error), and β = 0.80 (80% power) values were used in the G*Power 3.1.9.7 program, and hence we calculated that at least 16 participants were required for a parallel-design study. To account for potential dropouts, 32 participants were recruited. Inclusion criteria were as follows: (a) being healthy according to the Physical Activity Readiness Questionnaire (PAR-Q) [27], and (b) performing lower-limb exercises regularly. Exclusion criteria were as follows: (a) having any nervous/muscular disorder, (c) using stimulants, (d) using medication that would affect the physical performance before the tests, and (e) having any injury in the lower limbs over the last 6 months.

### 2.2. Study Design

All participants visited the laboratory for a total of three sessions and were divided into two groups: FR and PR. The determination of the groups was based on the selection of hidden cards offered to the participants by the researcher. Firstly, the height and weight of the participants were measured. The first session aimed to familiarize the participants with the tests (countermovement jump [CmJ], MVIC) and the exercise (NHE) and was also used to demonstrate how to apply the FR. During this familiarization session, all procedures were explained and demonstrated by the researcher, and participants were allowed to practice each test and exercise under supervision to ensure proper technique and understanding. In detail, for the NHE protocol (3 sets × 10 reps), the participants all used adjustments such as elastic bands or extra load, as needed, to achieve a standardized intensity of 17–20 on the Borg rate of perceived exertion (RPE) scale (scale 6–20) during the familiarization session. In the second session, all the participants started with a standardized warm-up, followed by the pre-test and then the NHE. Immediately after, participants either started a 5-min FR procedure or PR, which was then followed by the post-test_0h measurement. The post-test_48h measurement was then performed in the third session, after rest (48 h). Before the test sessions, participants performed a 5-min warm-up on a stationary bicycle (Monark, Ergomedic 874 E, Vansbro, Sweden) at 70 rpm⸱min^−1^ and 70 W. Each “test” in the study refers to the measurements of muscle stiffness, PPT, flexibility, MVIC torque, and CmJ height. Participants were instructed to avoid any workout for the 48 h before pre-test and between post-test_0h and post-test_48h. All the sessions were held at the same time of day, and the laboratory temperature was maintained at 22 °C. The NHE protocol was expected to induce DOMS (delayed onset muscle soreness), which was not considered an adverse effect but rather an intended and natural part of the study design. The study design is illustrated in Figure 1. Ethical approval was given by the ethical commission of University of Graz (approval code: GZ. 39/52/63 ex 2023/2024), all participants gave written informed consent prior to participation in the tests, and the protocol was conducted in accordance with the Declaration of Helsinki.

### 2.3. Procedures

#### 2.3.1. Tests

Muscle Stiffness Assessment (MSA): A MyotonPro device (Model: MyotonPro, Tartu, Estonia) was used to evaluate the muscle stiffness of the biceps femoris (BF) from the hamstring muscle group. Each participant was positioned in a prone position, with their hands flat on either side of their body and their feet naturally hanging off a physiotherapy bed. To determine the measuring point of the BF, a point was marked with a pen halfway (i.e., 50%) between the ischial tuberosity along the BF line and the fibular head [18,19]. The MyotonPro probe was held perpendicularly over the skin, and three consecutive impulses (0.42 N) assessed the deformation of the skin and muscle. Muscle stiffness was measured three times (nine impulses in total) and then averaged for analysis, as previously performed in studies [28,29]. According to a previous study, the intra-observer reliability of the MyotonPro has been reported as high, with an ICC of 0.89 and a coefficient of variation (CV) of 3.1% [30].

Pain Pressure Threshold (PPT): Algometric measurements were performed using a Pain Test FPX Algometer (Wagner Instruments, Riverside, CT, USA). PPT was defined in kg/cm^2^. The same spot as used for the muscle stiffness assessment was used in the PPT measurement [18,19]. Moreover, the same prone position as used in the muscle stiffness assessment was used for the PPT assessment, with the device always applied at a 90° angle to the skin surface. The participant was asked to say “STOP” when they felt the first obvious sensation of pain at the test point. The measurements were repeated three times, and the results were averaged. According to previous research, the intra-rater ICC ranged from 0.71 to 0.99, and the inter-rater ICC ranged from 0.81 to 0.86 [31].

Flexibility/Hip Flexion Range of Motion (ROM): Maximal hip flexion ROM was assessed using a sit and reach test, which was performed with a Sit n’ Reach Trunk Flexibility Box (New York, NY, USA). Each participant placed their feet firmly on the Sit n’ Reach box with their hips flexed and knees parallel and fully extended, with ankle joints in a neutral position (90°). In addition, the participant sat in an upright position with both arms parallel to the floor in front of their torso, with their index fingers touching each other. For the test procedure, the participant was asked to bend forward and move the stretch indicator on the Sit n’ Reach test box as far as possible with the fingertips of both hands. If any evasive movement with the legs or trunk was detected, the trial was repeated. The test was performed three times, with a 15-s break between trials, and the average of the three trials was taken for analysis [32]. The test–retest reliability of the Sit n’ Reach test has been reported in an earlier study as excellent (ICC = 0.95) [33].

Countermovement Jump (CmJ) Height: Each participant was asked to perform three CmJs on a portable force platform (Quattro Jump, Kistler GmbH, Winterthur, Switzerland, 500 Hz recording frequency), with a 1-min rest in between. The CmJ test protocol was used to assess jump height performance. A maximum jump was performed with the participant’s knees bent at an angle of their own choice during the countermovement, with hands fixed on the hips. Three trials were performed for each participant, and the trial with the highest jump height (in cm) was evaluated and used for the statistical analysis [32]. A previous study has shown the reliability of CMJ measurements to be excellent, with an ICC of 0.97 (95% CI: 0.91–0.99) [33].

Maximum Voluntary Isometric Contraction (MVIC) Peak Torque: Each participant was positioned on a dynamometer (Compass 540, Proxomed Group, Fürstenfeldbruck, Germany), with both hip and knee angles of 110° [34]. Consequently, the participant performed MVICs of the knee flexors unilaterally (the dominant leg used for kicking a ball). The center of rotation of the knee joint axis and the dynamometer were aligned with a custom-made laser device. The exact position of the participant during the initial MVICs was recorded to ensure the same position for all subsequent assessments on the dynamometer. To minimize evasive movements, the trunk and test leg were secured with straps, and the leg was fixed to the lever arm approximately 2 cm above the medial malleolus. With arms crossed in front of the chest, each participant performed three knee flexor MVICs of 5 s each, with a 1-min rest between trials. The participant pushed as hard as possible during the measurement, while the researcher provided verbal encouragement [32]. The mean of the three MVIC peak values was taken for further analysis. A previous study has shown the reliability of CMJ measurements to be excellent, with an ICC of 0.97 (95% CI: 0.91–0.99) [33].

Rate of Perceived Exertion (RPE): The Borg RPE scale (6–20) was used to determine the perceived difficulty of all participants after each NHE [35].

#### 2.3.2. Nordic Hamstring Exercise (NHE)

The NHE protocol began with each participant in a kneeling position, maintaining their torso upright and aligned above the knees. The participant’s heels and lower legs were securely fixed to a functional weight bench (Keiser, Lichtenau, Germany) using an adjustable ankle strap, ensuring that their feet remained in contact with the ground throughout the exercise (Figure 2a). The participant was instructed to use their hamstring muscles to resist the forward fall for as long as possible (e.g., 3–4 s) [2]. During the descent, the participant used their arms and hands to cushion the fall, allowing their chest to touch the surface of the functional weight bench. They then pushed themselves back to the starting position with their hands, thereby minimizing the load during the concentric phase. For participants performing NHE with extra load, a 20-centimeter-thick foam pad was placed on the bench surface to prevent hard impacts during the exercise (Figure 2c). In contrast, participants performing NHE with a resistance band used an elastic band that passed behind their chest (at neck level) and was anchored to a fixed metal bar behind them (Figure 2b). All the NHE protocols were standardized for the participants, as shown in Figure 2.

#### 2.3.3. Recovery

The recovery protocol was performed in two different ways, i.e., FR intervention and PR. In the FR intervention, a foam roller (Blackroll Standard Foam Rolling, Bottighofen, Switzerland) was used, and rhythmic movements from distal to proximal and from proximal to distal (every 2 s) of the posterior thigh (i.e., hamstring muscles) were performed, while the cadence was controlled by a metronome. The participant was asked to maintain a perceived pressure of 7/10 on a visual analog scale (VAS-10) [36]. Additional pressure on the leg was ensured by the researcher if the participant reported a sensation lower than 7 on the VAS-10 scale. The FR intervention consisted of two sets of 60 s (both legs, 240 s in total) for each leg muscle group [12,13]. For the PR group, the participant was asked to sit motionless on a chair for 240 s as a form of PR. The FR intervention is shown in Figure 3.

### 2.4. Statistical Analyses

The data were analyzed using IBM SPSS Statistics 27.0 software. Descriptive statistics are presented as mean and standard deviation. The Shapiro–Wilk test was used to assess normality. Baseline (pre-test) values between groups were compared using independent *t*-tests. For normally distributed data, a two-way repeated measures ANOVA was conducted to examine time and group interactions [two interventions (FR, PR) × three time points (pre-test, post-test_0h, post-test_48h)], and pairwise comparisons were made. For MVIC data, a logarithmic transformation was applied to achieve normal distribution prior to analysis. Effect sizes were calculated as partial eta square ($\eta_{p}^{2})$ for ANOVA. For $\eta_{p}^{2}$, values greater than 0.01, 0.06, and 0.14 were interpreted as “small”, “medium”, and “large” effect sizes, respectively. Bonferroni correction was applied for all the post hoc tests. The significance level was set as *p* < 0.05 for ANOVA.

## 3. Results

The anthropometrics of the participants are presented in Table 1. All baseline (pre-test) measurements, including stiffness, MVIC, flexibility, and pain threshold, were compared between the FR and PR groups using independent samples t-tests. No statistically significant differences were found (all *p* > 0.05), indicating baseline equivalence. The mean values, pairwise comparisons, and main effects of the time and group–time interactions of the tests performed at the three different time points (pre-test, post-test_0h, and post-test_48h) are presented in Table 2.

### 3.1. Muscle Stiffness

In the analysis for muscle stiffness, the main effect of the time and group interaction was not statistically significant (F_2,60_ = 0.155, *p* = 0.858, $\eta_{p}^{2}$ = 0.022) indicating a small interaction effect. However, the main effect of time was significant for muscle stiffness changes (F_2,60_ = 27.709, *p* = 0.001, $\eta_{p}^{2}$ = 0.798), reflecting a large time effect. This indicates that both groups showed a similar change in muscle stiffness. Specifically, in the FR group, a significant increase of 6% was observed between the pre-test and post-test_0h measurements (*p* < 0.05), while the PR group showed a 5.5% increase. For the FR group, there was a significant decrease from pre-test to post-test_48h (*p* = 0.015, Figure 4). In contrast, no significant difference was found between the pre-test and post-test_48h measurements in the PR group (*p* > 0.05). In addition, there was no significant difference between the post-test_0h and post-test_48h measurements in either group (*p* > 0.05).

### 3.2. Pain Pressure Threshold (PPT)

For PPT, the main effect of the time and group interaction was not significant (F_2,60_ = 0.474, *p* = 0.632, $\eta_{p}^{2}$ = 0.063) indicating a medium interaction effect and no group-based differences. However, the main effect of time was significant for PPT changes (F_2,60_ = 20.519, *p* = 0.001, $\eta_{p}^{2}$ = 0.746) pointing to a large interaction effect. Both groups showed no significant difference between the pre-test and post-test_0h measurements (*p* > 0.05). In the PR group, a significant decrease of 14.9% was observed from pre-test to post-test_48h (*p* = 0.012, Figure 4), whereas no significant change was found in the FR group during the same period (*p* > 0.05). Between post-test_0h and post-test_48h, the FR group demonstrated a significant decrease of 14.2% (*p* < 0.001), and the PR group exhibited an 18.1% decrease (*p* = 0.04, Figure 4).

### 3.3. Flexibility

The interaction effect between time and group for the flexibility parameter was statistically significant (F_2,60_ = 16.141, *p* = 0.001, $\eta_{p}^{2}$ = 0.834) confirming a large effect of time. In addition, the main effect of time was also significant (F_2,60_ = 48.401, *p* = 0.001, $\eta_{p}^{2}$ = 0.874) indicating a large interaction effect. In the FR group, flexibility increased significantly by 12.7% from pre-test to post-test_0h (*p* < 0.001), whereas no such change was observed in the PR group (*p* > 0.05). A significant 5.3% decrease in flexibility was recorded in the PR group from pre-test to post-test_48h (*p* = 0.011), but this reduction was not observed in the FR group (Figure 4). Between post-test_0h and post-test_48h, both groups exhibited a significant decrease in flexibility: 8.8% in the FR group (*p* < 0.001) and 9.2% in the PR group (*p* = 0.004, Figure 4).

### 3.4. Countermovement Jump (CmJ) Height

For the CmJ height values, the interaction between group and time was not significant (F_2,60_ = 1.636, *p* = 0.232, $\eta_{p}^{2}$ = 0.189) reflecting a large interaction effect. However, the main effect of time was significant (F_2,60_ = 9.305, *p* = 0.003, $\eta_{p}^{2}$ = 0.571) indicating a large effect. No significant change was observed in CmJ height performance from pre-test to post-test_0h in either group (*p* > 0.05). In the FR group, a 3.3% increase was observed from pre-test to post-test_48h (*p* = 0.017, Figure 4), while no significant change was found in the PR group during the same period (*p* > 0.05). From post-test_0h to post-test_48h, the FR group showed a significant 3% increase in jump performance (*p* = 0.019), and the PR group demonstrated a 3.4% increase during the same period (*p* = 0.012, Figure 4).

### 3.5. Maximal Voluntary Isometric Contraction (MVIC) Peak Torque

The interaction effect between time and group for the MVIC peak torque values parameter was not statistically significant (F_2,60_ = 0.791, *p* = 0.473, $\eta_{p}^{2}$ = 0.102) demonstrating a medium effect. However, the main effect of time was significant (F_2,60_ = 6.636, *p* = 0.011, $\eta_{p}^{2}$ = 0.475) indicating a large effect. In the PR group, a significant decrease of 7.8% was observed from pre-test to post-test_0h (*p* = 0.001), while no significant change occurred in the FR group during this period (*p* > 0.05, Figure 4). From pre-test to post-test_48h, there was a significant decrease of 7.1% for the PR group (*p* = 0.047); however, there was no significant difference for the FR group (*p* > 0.05, Figure 4). In addition, no significant changes were observed in either group from post-test_0h to post-test_48h (*p* > 0.05).

## 4. Discussion

The aim of this study was to investigate the effects (i.e., acutely and after 48 h) of an FR intervention on parameters such as muscle stiffness, PPT, flexibility, CmJ height, and MVIC peak torque following the NHE. The main findings showed that FR induced significant improvements in flexibility immediately after the NHE, compared to PR, while no such changes were observed in muscle stiffness, PPT, CmJ height, and MVIC peak torque. At the 48-h mark, FR showed no significant benefits in terms of muscle stiffness, PPT, CmJ height, or MVIC peak torque, similar to PR; however, FR was in favor, compared to PR, in flexibility.

Concerning flexibility, the FR intervention showed an immediate improvement after the NHE, compared to the PR group (see Table 2). This finding is supported by many studies [20,32,37]. By increasing muscle relaxation and blood flow, FR allows muscles to be held in longer positions, which can lead to increased flexibility [20]. In addition, the acute improvement in flexibility following FR may be related to several mechanisms, including a reduction in muscle stiffness, increased stretch tolerance, and thixotropic effects [11]. More precisely, studies have suggested that FR can decrease muscle stiffness, thereby enhancing ROM [7,12]. In addition, increased stretch tolerance, related to changes in pain perception, can contribute to the prolonged ROM even after the cessation of structural changes [38]. Thixotropic effects, involving reduced fluid viscosity in the treated muscle, skin, and fascia, can also reduce resistance to movement and facilitate flexibility gains [12]. In contrast, the PR group showed a significant decrease in flexibility at 48 h, compared to baseline, whereas no such decrease was observed in the FR group. These findings suggest that while FR may offer short-term benefits for flexibility immediately after the NHE, these effects do not persist after 48 h. However, the immediate improvement in flexibility may indicate potential recovery processes. In addition, the FR intervention can help prevent the potential loss of flexibility due to a reduced DOMS sensation, particularly after strenuous eccentric exercise (e.g., the NHE) [15]. Similarly, Kasahara et al. [18] presented findings that suggested that FR can preserve flexibility by accelerating muscle recovery.

Moreover, for the results on muscle stiffness, there was no significant difference between the two groups. However, the main effect of time for muscle stiffness was significant, indicating that neither method was more effective in reducing muscle stiffness. In recent meta-analyses, neurological (stretch tolerance) and structural (muscle stiffness) parameters have been evaluated [14,16]. FR is a soft tissue massage technique aimed at reducing tension caused by inelastic or adhesive fascia, especially in injury or pathological conditions [39]. Although it is not clear whether FR releases myofascia [12], acute increases in soft tissue flexibility, pain thresholds, and stretch tolerance have been reported [5]. Furthermore, it has been suggested that the change in pain perception (rather than the change in muscle stiffness) is the mechanism for increases in ROM [16]. Moreover, although a decrease in muscle stiffness has the potential to reduce injury risk, this decrease could lead to a short-term decrease in force production [40], at least following an acute bout of the NHE. Similarly, Kasahara et al. [18] reported that FR did not provide the expected effect on muscle stiffness after eccentric exercises. Other studies have indicated that the possible decrease in muscle stiffness can have negative effects on muscle performance in the short term by delaying the healing of micro-tears caused by eccentric exercises [3,13]. In this context, this discrepancy suggests that the mechanisms of flexibility and muscle stiffness could be different and that this increase in flexibility might be explained by another mechanism (e.g., stretch tolerance). In addition, FR may increase flexibility by increasing muscle relaxation and blood flow, but its effect on muscle stiffness may be limited.

Regarding PPT, time effects only were shown, and these changes were likely explained by the NHE-induced DOMS in both groups [41]. After the NHE, both groups showed a slight increase in PPT, compared to baseline values, but this increase was not significant. This could likely be related to neural adaptations, as during eccentric exercises, temporary desensitization of mechanoreceptors and the suppression of pain pathways by the central nervous system can lead to a short-term increase in PPT [42]. The effects of active recovery methods, such as FR, on PPT suggest that neural adaptations modulate post-exercise pain perception and that metabolic by-products in the muscles can create a temporary pain-dampening effect [43]. In this context, the observed increase in PPT following eccentric exercise can be attributed to neural and mechanical adaptations, whereas the increase in muscle stiffness can be considered a physiological response due to microtrauma and inflammation. This indicates that pain perception is regulated by neural and mechanical adaptations, while muscle stiffness is more closely linked to tissue damage and inflammation. These findings suggest that different physiological mechanisms could be simultaneously at play and could balance each other out. In addition, in the PR group, a significant decrease in PPT was observed at the 48-h measurement, compared to baseline, whereas no such decrease was observed in the FR group. In line with this, Pearcey et al. [15] reported that FR resulted in less of a decrease in muscle PPT at the 24-h and 48-h marks, compared to PR, after intense exercise. DOMS is thought to be caused by both muscle and connective tissue damage, but connective tissue damage appears to play a more prominent role [44]. Specifically, inflammation in the connective tissue and the accumulation of cells and fluid in interstitial spaces contribute to the sensation of pain. FR’s ability to increase blood flow, reduce connective tissue damage, and alleviate pain through mechanically sensitive receptors can explain its effectiveness. Although significant decreases in PPT were observed in both groups between the immediate post-NHE measurement and 48 h later, this decrease was more limited in the FR group than in the PR group. These findings suggest that FR may be a more effective strategy than PR for recovery from connective tissue damage and inflammation.

The findings related to CmJ height revealed that there were significant increases in the jump performance of the participants over time, but these increases did not show a significant difference between groups. After the NHE, no significant improvement in jump performance was observed in either group, compared to baseline. One possible reason for this is that muscle groups such as the quadriceps, gluteus maximus, and gastrocnemius play a dominant role during the upward thrust phase of the CmJ, while the hamstrings are less involved [45]. In contrast, the NHE primarily targets the hamstrings and, as a result, the hamstrings could have been fatigued following the NHE, but the quadriceps were likely unaffected, compared to the hamstrings. This could explain why jump performance did not change significantly despite hamstring fatigue following NHE. The improvement in jump performance observed after 48 h may be related to the learning effect and the passage of time. Repeated testing helps participants develop more efficient movement patterns and better utilize elastic energy [46]. In addition, fatigue caused by eccentric loading typically decreases within 48 h, allowing muscle function to recover over time. These factors together could explain the improvements in CmJ performance for both groups, although there was no significant difference following the interventions.

Concerning MVIC peak torque, the main effect of time was significant, but the FR intervention had no additional effect on MVIC, compared to PR. Following the NHE, although a decrease in MVIC values was observed in the FR group, compared to baseline, this change was not statistically significant. On the other hand, the decrease was significant in the PR group. This may be explained by factors such as natural fatigue, the accumulation of acidic metabolites, and reduced muscle function following maximal force exertion in the hamstring muscles [13]. In addition, microtraumas induced by eccentric exercises such as the NHE can hinder muscles from utilizing their full potential during isometric contractions [47]. Consequently, acute fatigue can temporarily reduce muscle function, leading to performance loss. Moreover, in both groups, MVIC values continued to decline at 48 h, indicating that the effects of DOMS induced by the NHE persisted, despite the interventions. The lack of a significant FR effect may be due to its passive nature, which may not provide sufficient neuromuscular stimulation to counteract fatigue or restore voluntary activation following eccentric exercise [48]. Previous studies have also reported conflicting findings. Sullivan et al. [49] noted that FR had no effect on MVIC in the absence of strenuous exercise, while Reiner et al. [7] reported that FR increased MVIC performance (+8.6 Nm). This study concluded that the FR intervention, when combined with a demanding exercise such as the NHE, is ineffective in improving MVIC acutely or at 48 h post-exercise.

Interestingly, CmJ height increased despite a decrease in MVIC peak torque. While MVIC assesses the maximum isometric force production capacity of muscles, the stretch-shortening cycle (SSC) mechanism utilized during CmJ enables the storage and release of elastic energy within the muscle-tendon complex. This mechanism likely supports the maintenance or even enhancement of performance under fatigue conditions [50]. In contrast, MVIC assessments evaluate only the direct force generated by muscles. Fatigue or microtraumas induced by the NHE can directly impair MVIC [47]; however, this impairment does not necessarily extend to jump performance. Moreover, jump performance involves not only physiological aspects but also technical skills that can improve through learning effects [46]. Conversely, MVIC remains unaffected by technical improvements and is more susceptible to neuromuscular fatigue. Following intense resistance exercises, neuromuscular fatigue can limit the muscles’ capacity to perform maximally during MVIC assessments. As a result, MVIC values can decline after eccentric exercises [13], whereas CmJ performance compensates for fatigue through the elastic energy provided by the SSC and rapid muscle contractions.

In conclusion, our study observed that FR intervention following NHE increased flexibility but showed no significant effect on other muscle parameters such as MVIC, CMJ, PPT, and muscle stiffness. It is likely that the improvement in flexibility stems from neurological adaptations such as increased pain tolerance and muscle relaxation, while muscle performance and stiffness are more related to mechanical properties and motor unit activation [13,15,18]. Therefore, it is possible that the effect of FR on these mechanical and performance-related parameters is limited. Additionally, the lack of change in CMJ performance may be due to FR being applied only to the hamstrings, excluding dominant muscle groups [45]. The limited change in PPT may be attributed to FR’s superficial application, which may not sufficiently reduce deep tissue inflammation and pain. These findings suggest that recovery involves multiple physiological mechanisms and that considering individual variability is important for optimizing recovery strategies.

This study has several limitations. Firstly, it was conducted with male participants only, which limits the generalizability of the findings. Future research should investigate gender differences by including both male and female participants. Moreover, the randomization method (card draw) may have introduced bias due to inadequate allocation concealment; thus, future studies should use software-generated randomization. In this study, the VAS scale was only used to standardize pressure in the FR intervention, but it could also be used to assess DOMS after the NHE. Additionally, no other objective measures were used to standardize NHE intensity or FR pressure. Furthermore, outcome assessors were not blinded, and because the PR (sitting) condition did not match the tactile/mechanical stimulation of FR, a placebo effect cannot be ruled out, which may have introduced expectancy bias. This study examined the acute effects immediately after strenuous exercise and 48 h later and does not provide information on long-term recovery processes. Future research could provide more comprehensive results by also investigating long-term recovery effects. In addition, the study only evaluated the FR intervention, compared to PR. Comparing different recovery methods could help us to better understand the recovery process.

## 5. Conclusions

The hypotheses of this study were partially supported. According to the results of the study, FR after the NHE had positive acute effects on flexibility, compared to PR, as well as a favorable effect after 48 h; however, its effects on PPT, muscle stiffness, CmJ height, and MVIC peak torque were limited both acutely and after 48 h. In conclusion, FR may serve as a practical and time-efficient strategy to enhance flexibility following strenuous eccentric exercise, though its effects on pain sensitivity, muscle stiffness, and strength recovery remain limited. Nevertheless, factors such as application protocols and individual differences can play a critical role in its effectiveness.

## Figures and Tables

**Figure 1 medicina-61-01486-f001:**
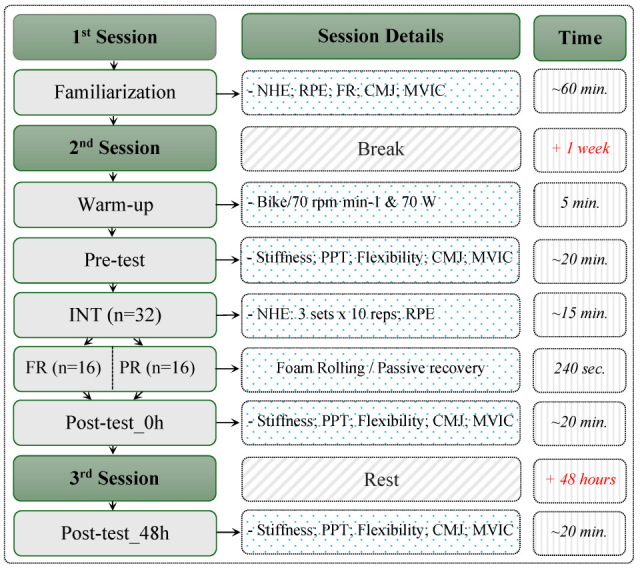
Design of the study.

**Figure 2 medicina-61-01486-f002:**
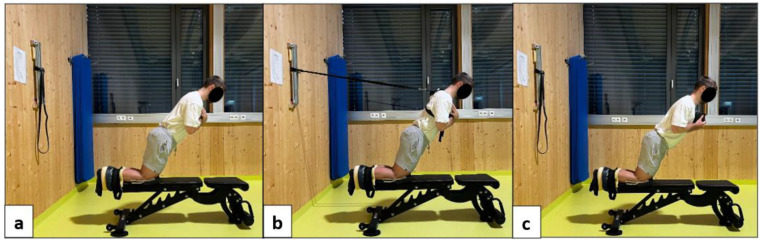
Nordic hamstring exercise protocols. (**a**) NHE; (**b**) NHE with elastic band; and (**c**) NHE with extra load.

**Figure 3 medicina-61-01486-f003:**
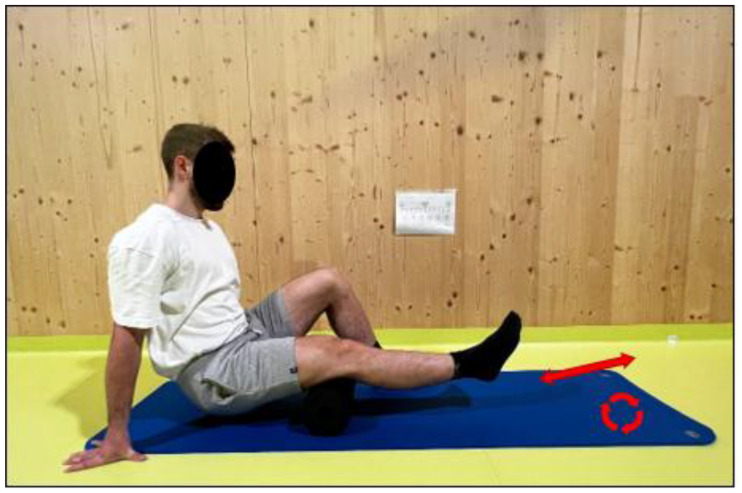
The foam rolling intervention. The red arrows in the figure illustrate the rhythmic back-and-forth motion of the foam roller, moving from the lower (distal) to the upper (proximal) part of the thigh and back again. The red circles highlight the rolling action along the muscle, emphasizing the specific areas targeted during foam rolling.

**Figure 4 medicina-61-01486-f004:**
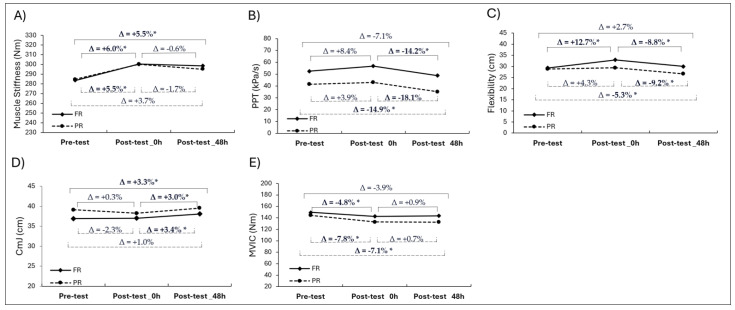
Repeated measures ANOVA results showing within-group time effects on muscle function variables: (**A**) muscle stiffness; (**B**) PPT; (**C**) flexibility; (**D**) CMJ; (**E**) MVIC at pre-test, post-test_0h, and post-test_48h (Δ = change %) (* Significant difference, *p* < 0.05).

**Table 1 medicina-61-01486-t001:** Descriptive statistics of the participants.

Variable	FR			PR		
	Mean (SD)	Min	Max	Mean (SD)	Min	Max
Age (years)	23.7 (3.7)	18	29	22.8 (3.5)	18	29
Height (cm)	178.2 (5.3)	170	192	179.7 (6.7)	168	195
Weight (kg)	75.0 (10.3)	52	91	75.7 (13.1)	60	114
BMI (kg/m^2^)	21.0 (2.7)	15	24	21.0 (3.2)	16	30
RPE	18.0 (0.8)	17	19	18.2 (0.8)	17	20
PA (MET-min/wk)	3997.3 (2070.1)	1272	9378	4462.8 (2332.7)	1775	10.344

Note: FR: foam rolling; PR: passive recovery; PA: physical activity; RPE: rate of perceived exertion; kg: kilograms; cm: centimeter; min: minimum; max: maximum; SD: standard deviation; BMI: body mass index.

**Table 2 medicina-61-01486-t002:** Means, confidence intervals, and post hoc comparisons of the muscle stiffness, PPT, flexibility, CmJ height, and MVIC parameters.

Variable	Time	FR		PR	
		Mean (SD)	[95% CI]	Mean (SD)	[95% CI]
Stiffness (Nm)	Pre-test	283.5 (32.1) ^ab^	[266.4–300.1]	284.6 (37.3) ^a^	[264.7–304.6]
	Post-test_0h	300.4 (29.6)	[284.6–316.2]	300.2 (49.2)	[274.0–326.4]
	Post-test_48h	298.6 (27.3)	[284.0–313.1]	295.2 (39.8)	[274.0–316.4]
PPT (kPa/s)	Pre-test	52.5 (19.1)	[42.3–62.7]	41.5 (14.0) ^b^	[34.0–49.0]
	Post-test_0h	56.9 (17.5) ^b^	[47.6–66.2]	43.1 (15.2) ^b^	[35.0–51.2]
	Post-test_48h	48.8 (17.7)	[39.4–58.2]	35.3 (13.7)	[28.0–42.6]
Flexibility (cm)	Pre-test	29.2 (8.9) ^a^	[24.5–34.0]	28.2 (8.2) ^b^	[24.4–33.2]
	Post-test_0h	32.9 (8.2) ^b^	[28.6–37.3]	29.4 (8.1) ^b^	[25.1–33.6]
	Post-test_48h	30.0 (8.8)	[25.3–34.6]	26.7 (8.6)	[22.1–31.2]
CmJ (cm)	Pre-test	36.9 (4.3) ^b^	[34.4–39.4]	39.1 (5.4)	[36.6–41.6]
	Post-test_0h	37.0 (4.1) ^b^	[34.7–39.3]	38.2 (4.7) ^b^	[36.0–40.6]
	Post-test_48h	38.1 (4.0)	[35.7–40.2]	39.5 (5.1)	[37.2–41.9]
MVIC (Nm)	Pre-test	149.6 (30.6)	[133.3–166.0]	144.2 (32.5) ^ab^	[129.5–159.2]
	Post-test_0h	142.4 (29.0)	[127.0–157.9]	133.0 (33.7)	[117.0–148.8]
	Post-test_48h	143.7 (30.6)	[127.2–160.0]	134.0 (30.1)	[120.6–148.1]

Note: FR: foam rolling; PR: passive recovery; CI: confidence interval; post hoc comparisons: (a: significantly different from post-test_0H (*p* < 0.05); b: significantly different from post-test_48h (*p* < 0.05)).

## Data Availability

The data analyzed during the current study are accessible from the corresponding author on reasonable request.

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
