# Peer review of "A Single Bout of Foam Rolling After Nordic Hamstring Exercise Improves Flexibility but Has No Effect on Muscle Stiffness or Functional Muscle Parameters"

_medicina, 2025, doi:10.3390/medicina61081486_

Round 1

Reviewer 1 Report

Comments and Suggestions for Authors

see attached

Comments on the Quality of English Language

Manuscript needs review for wordiness, use of different terms to express the same idea, and readability. I strongly recommend that a trusted colleague not familiar with the study provide a thorough editing. Those familiar with the study know what they are looking for, whereas someone external will recognize where the writing can be improved.

Reviewer 2 Report

Comments and Suggestions for Authors

This study aims to assess foam rolling as a potential strategy to counteract fatigue following the Nordic hamstring exercise. This manuscript is intriguing, but it suffers from some methodological flaws and overinterpretation of a narrow effect.  More detailed suggestions are presented below:

Introduction:

  1. Overall, the introduction is long and hard to read. It would benefit from more concise paragraphing. The first 3–4 paragraphs could be cut by 30–40% without losing meaning. Try focusing on FR after NHE (which is the actual research gap).
  2. Introduction lacks a focused rationale for selecting only foam rolling vs. other recovery modalities (e.g., active recovery, stretching, vibration, etc.).
  3. Hypotheses are stated, but not specific enough in terms of expected magnitudes or directions of change across all measured outcomes. Try being explicit in the hypothesis: e.g., “FR will prevent decreases in MVIC and flexibility, and reduce muscle stiffness and pain.”. Of course, hypotheses should be linked to the previous findings.

Methods:

  1. The randomisation process is inadequate. Card draw is not a valid allocation concealment technique; this introduces bias. A proper random allocation sequence (e.g., software-generated) should be used.
  2. This study's methodology lacks blinding of outcome assessors, a significant flaw. Moreover, there is no control for the placebo effect: PR (sitting) does not match the tactile/mechanical stimulation of FR, so expectancy bias is not ruled out. Please thoroughly elaborate on this and include it in the limitations.
  3. FR dose/duration (240 sec) is arbitrarily chosen. No justification or reference was provided. Similarly, no reliable or valid data are provided for the measurement protocols.
  4. Mixing Two-way ANOVA with non-parametric tests in a single study is a questionable methodology that limits the findings. Usually, if you have a non-normal distribution, some other data transformation techniques are first conducted (prior to resolving the non-parametric tests).

Results:

  1. There is a lack of a baseline comparability table: Were the FR and PR groups equivalent in pre-test values across all variables? Was this tested with ANOVA? If there are no statistically significant differences, this should be explicitly mentioned.
  2. Table 2 is too dense. Confidence intervals are reported, but effect sizes (ηp²) are not discussed or interpreted. Please include plots (e.g., line graphs) to show trends over time for each variable and analyse the effect size as mentioned in the methods.

Discussion:

  1. The discussion is too long and hard to read. It also overinterprets flexibility findings as evidence of “recovery” without demonstrating return to baseline or performance enhancement.
  2. Some explanations are speculative, especially regarding neural mechanisms, with no EMG or imaging support. Also, there is no in-depth discussion on the lack of effect on performance variables (e.g., why FR didn’t help MVIC or CmJ). Try expanding the discussion of individual vs. group responses and practical implications for athletes
  3. List all major limitations: male-only sample, unblinded design, possible placebo effect, no long-term follow-up.

Reviewer 3 Report

Comments and Suggestions for Authors

General Comment

This manuscript presents a well-structured and engaging study that compares the effects of foam rolling (FR) versus passive recovery (PR) following the Nordic Hamstring Exercise (NHE). The authors assess a broad range of physiological and perceptual outcomes including flexibility, muscle stiffness, pain pressure threshold (PPT), countermovement jump (CmJ) height, and maximal voluntary isometric contraction (MVIC). The methodology is generally sound, and the findings are relevant to sports scientists, clinicians, and practitioners seeking effective post-exercise recovery strategies. However, some areas require further clarification or refinement to enhance clarity, reproducibility, and interpretation. These improvements will strengthen the impact and transparency of the study.

Detailed Comments

  1. Introduction
  • The introduction provides a solid foundation for the study. To enhance clarity and originality, explicitly state the specific gap in the existing literature. For example, emphasize that few studies have examined both the immediate and delayed effects of FR specifically following the NHE on functional and perceptual recovery markers.
  1. Methods
  • Clarify the rationale for using non-parametric statistical methods (e.g., Kruskal-Wallis test) for certain variables. A brief statement confirming non-normality (e.g., based on the Shapiro-Wilk test) would support this choice and improve transparency.
  • Include more specific details regarding the FR protocol to allow replication. Indicate the duration per muscle group, total rolling time, number of sets, and any instructions related to applied pressure or discomfort levels. These factors are essential for reproducibility and practical application.
  1. Results
  • The results are comprehensive and well organized. However, several explanatory or interpretive statements (e.g., “due to DOMS” or “likely due to...”) appear in this section. These should be relocated to the Discussion, in line with standard reporting conventions.
  • Table 2 is informative but dense. Highlight statistically significant results (e.g., via boldface or underlining) to aid reader comprehension. Additionally, ensure the post-hoc notations (e.g., “a” and “b”) are clearly explained in the table legend for clarity.
  1. Discussion
  • The discussion of acute improvements in flexibility is thorough and well-supported by previous literature. However, when proposing mechanisms (e.g., stretch tolerance, thixotropic effects, fascial changes), acknowledge that these were not directly measured. Frame these mechanisms as potential explanations rather than definitive conclusions.
  • The contrast between CmJ and MVIC results is well-articulated and highlights the difference between functional movement and isolated strength post-eccentric loading. This discussion is a key strength of the manuscript.
  • The section on muscle stiffness is somewhat repetitive. Consolidating related paragraphs and clearly stating whether the observed changes are beneficial, neutral, or potentially maladaptive will improve the flow and clarity of this section.
  1. Limitations
  • The study's limitations are appropriately addressed, including the male-only sample and short (48-hour) follow-up period. Two additional limitations should be acknowledged:
    • The absence of post-NHE DOMS ratings (e.g., via a visual analog scale) limits the interpretation of perceptual recovery and its relation to PPT.
    • The lack of a no-treatment control group restricts the ability to determine the natural recovery trajectory. Consider adding a sentence such as: “The absence of a no-treatment control group limits our ability to isolate the natural time-course of recovery independent of any intervention.”
  1. Conclusion
  • The conclusion is consistent with the results, though somewhat conservative. To enhance its practical relevance, consider stating:
    “Foam rolling may serve as a practical and time-efficient strategy to enhance flexibility and reduce pain sensitivity following strenuous eccentric exercise, though its effects on muscle stiffness and strength recovery remain limited.”

Round 2

Reviewer 1 Report

Comments and Suggestions for Authors

I appreciate the authors' attention to detail in addressing my comments.

Reviewer 2 Report

Comments and Suggestions for Authors

Dear authors,

I appreciate the effort you took to address all the issues and comments and truly improve this manuscript. I believe it is now suitable for publication.

Best regards